# Solving a North-type energy balance model using boundary integral methods

Aksel Samuelsberg[1] and Per Kristen Jakobsen[1]

[1]Department of Mathematics and Statistics, UiT - The Arctic University of Norway, Tromsø, Norway

**Correspondence:** Aksel Samuelsberg (aksel.samuelsberg@uit.no)

**Abstract.** Simplified climate models, such as energy balance models (EBMs) are useful conceptual tools, in part because their reduced complexity often allows for studies using analytical methods. In this paper, we solve a North-type EBM using a boundary integral method (BIM). The North-type EBM is a diffusive, one-dimensional EBM with a nonlinear albedo feedback mechanism. We discuss this approach in light of existing analytical techniques for this type of equation. Subsequently, we test the proposed method by solving multiple North-type EBMs with a zonally symmetric continent featuring an altered ice-albedo feedback dynamic. We demonstrate that the introduction of a continent results in new equilibrium states characterized by multiple ice edges and ice belts. Furthermore, we show that the BIM serves as an efficient framework for handling unconventional ice distributions and model configurations for North-type EBMs.

## 1 Introduction

Despite the advancement in computational power, conceptual climate models remain valuable tools for understanding the Earth's climate system. The complexity of realistic models has highlighted the need for a hierarchical model structure where conceptual models provide a solid theoretical foundation as model complexity increases (Schneider and Dickinson, 1974; Claussen et al., 2002; McGuffie and Henderson-Sellers, 2014). Energy balance models (EBMs) stand out as some of the simplest climate models. Their simplicity allows for both analytical and numerical studies of climate responses to forcings. So-called zero-dimensional EBMs describe the Earth's global mean temperature and include no spatial variables. Zero-dimensional EBMs are readily examined using analytical tools (North, 1990; Ghil and Lucarini, 2020; Lohmann, 2020). One-dimensional EBMs with a latitude dependence, coupled with a linear transport term and a nonlinear albedo feedback, often called Budyko-type models, also lend themselves to analytical investigations (Budyko, 1969; Held and Suarez, 1974; Widiasih, 2013; Walsh and Widiasih, 2014). A more physically motivated transport mechanism (Rose and Marshall, 2009) may be included in the model by instead adding a diffusion term. One-dimensional EBMs with meridional heat transport by diffusion, hereafter called North-type models, have attracted considerable interest (North et al., 1981; Ghil, 1976; Bódai et al., 2015; Del Sarto et al., 2024). While the inclusion of the diffusion term complicates the models, mathematically speaking, in certain configurations these models may be studied using analytical methods. Pioneering analytical investigations into these models have been conducted by North (North, 1975a, b; North et al., 1981). A general solution expressed through a Fourier-Legendre series for the equilibrium temperature field was found using spectral methods for a step function albedo. Although this solution is rapidly

converging for standard model configurations such as the idealized aquaplanet, a geographical input with parameter discontinuities across land-sea boundaries causes the spectral solution to converge slowly (Mengel et al., 1988; North and Kim, 2017).

In this paper, we solve the stationary form of an EBM with a meridional heat transport and a nonlinear albedo feedback using an analytical method: the boundary integral method (BIM). This North-type EBM describes the zonal mean surface temperature with its key features being a linear heat diffusion across latitudes, an ice-albedo feedback mechanism, as well as the stabilizing effect of the outgoing longwave radiation. The model may be formulated as

$$C\frac{\partial T}{\partial t} - D\frac{1}{\sin\theta}\frac{\partial}{\partial\theta}\Big(\sin\theta\frac{\partial T}{\partial\theta}\Big) + BT = Qs(\theta)(1 - a(T)) - A \tag{1}$$

using spherical coordinates, where the polar angle, $\theta$, is the latitude. The latitude ranges from $\theta = 0$, the North Pole, to $\theta = \pi$, the South Pole. Here $C$ represents the heat capacity of the lower atmosphere and hydrosphere with an assigned average constant value, and $D$ is the diffusion rate determining the strength of the meridional heat transport. The parameter $Q$ is defined as one-fourth of the mean total solar irradiance (TSI), as the disk silhouette capturing solar radiation is one-fourth of the Earth's total area. To address the solar radiation distribution across latitudes, the model incorporates an average annual latitudinal energy distribution function, denoted $s(\theta)$. Additionally, the model assumes a constant lapse rate, establishing a linear relationship between surface temperature and outgoing energy, expressed as $E_{\text{out}} = A + BT$, where $A$ and $B$ are constants (Budyko, 1969). The ice-albedo feedback mechanism is included by allowing for a temperature dependent albedo,

$$a(T) = \begin{cases} a_1, & T > -T_s \\ a_2, & T < -T_s, \end{cases} \tag{2}$$

where $-T_s$ is the critical temperature for ice formation at the surface. Latitudes with an annual mean temperature below $-T_s$ are deemed to have an ice cover. Consequently, there will be a critical latitude at which the ice cover ends and begins. A major challenge arises in determining the location of the critical latitude under the given constraints. We show that the BIM offers a convenient way to address this, even for scenarios with several critical latitudes.

The proposed method is tested on a model configuration where the idealized aquaplanet is given a zonally symmetric continent with an altered ice-albedo feedback mechanism. Equilibrium solutions to Eq. (1) are found and a bifurcation diagram is drawn for three different systems with a zonally symmetric continent. Lin and North (1990) previously studied similar zonal band continent configurations and a circular cap of land centered at one pole was studied by Mengel et al. (1988). In these studies, land and sea were differentiated by a change in heat capacity, with no change in the stationary equation. Here, we demonstrate that the BIM represents an analytical method that efficiently handles arbitrary parameter discontinuities at the land-sea interface in North-type EBMs. Additionally, we show that the introduction of a continent with altered equilibrium parameters gives rise to new equilibrium states, characterized by unconventional ice distributions featuring multiple ice edges and ice belts.

## 2 Results

BIMs are a general approach for boundary value problems, where the problem is reduced to boundary integral equations involving an associated Green's function (Hsiao and Wendland, 2008; Morino and Piva, 2012). To showcase the application of the BIM in the context of EBMs, we employ it to find the equilibrium solutions to the classical idealized aquaplanet. Using $T_s$ as a scale for temperature, $T = T_s \tilde{T}$, where $\tilde{T} = \tilde{T}(\theta)$ is the non-dimensional temperature field at latitude $\theta$, we may write the stationary form of the energy balance equation (1) in non-dimensional form;

$$-\frac{1}{\sin\theta}\frac{\partial}{\partial\theta}\left(\sin\theta\frac{\partial\tilde{T}}{\partial\theta}\right) + \beta\tilde{T} = \eta s(\theta)(1 - a(\tilde{T})) - \alpha, \tag{3}$$

where $\beta = \frac{B}{D}$, $\alpha = \frac{A}{T_s D}$ and $\eta = \frac{Q}{T_s D}$. Hereafter, the tilde notation on the non-dimensional temperatures will be omitted, as the subsequent analysis will focus exclusively on non-dimensional temperatures. Defining

$$\mathcal{L}(\cdot) = -\frac{1}{\sin\theta}\frac{\partial}{\partial\theta}\left(\sin\theta\frac{\partial}{\partial\theta}(\cdot)\right) + \beta(\cdot), \tag{4}$$

it can be shown that, for two functions $v$ and $u$ on the domain $[\theta_1, \theta_2]$, we have

$$\int_{\theta_1}^{\theta_2} d\theta\,\sin\theta\{v\mathcal{L}u - u\mathcal{L}v\} = \left\{u\sin\theta\frac{\partial v}{\partial\theta} - v\sin\theta\frac{\partial u}{\partial\theta}\right\}\Big|_{\theta_1}^{\theta_2}. \tag{5}$$

Defining

$$h(T,\theta) = \eta s(\theta)(1 - a(T)) - \alpha, \tag{6}$$

we may write Eq. (3) in the compact form

$$\mathcal{L}T = h. \tag{7}$$

Let $K$ be a Green's function for the operator (4). That is,

$$\mathcal{L}K(\theta,\xi) = \delta_\xi(\theta), \tag{8}$$

where $\delta_\xi(\theta)$ is a Dirac-delta function along the curve $\theta \in [0, \pi]$ (see Appendix A, Eq. (A4)). Inserting $v = T$ and $u = K$ in the identity (5) we have

$$\int_{\theta_1}^{\theta_2} d\theta\,\sin\theta\left\{T\mathcal{L}K - K\mathcal{L}T\right\} = \left\{K\sin\theta\frac{\partial T}{\partial\theta} - T\sin\theta\frac{\partial K}{\partial\theta}\right\}\Big|_{\theta_1}^{\theta_2}. \tag{9}$$

Inserting Eq. (7) and Eq. (8) into Eq. (9) and rearranging for $T$, any exact solution $T(\theta)$ to Eq. (3) must satisfy the identity

$$T(\xi) = \int_{\theta_1}^{\theta_2} d\theta\,\sin\theta\,K(\theta,\xi)h(T,\theta) + \left\{K\sin\theta\frac{\partial T}{\partial\theta} - T\sin\theta\frac{\partial K}{\partial\theta}\right\}\Big|_{\theta_1}^{\theta_2}. \tag{10}$$

A suitable Green's function is found in Appendix A,

$$
K(\theta,\xi) = \begin{cases} \dfrac{P_\lambda(\cos\xi)(\pi\cot(\pi\lambda)P_\lambda(\cos\theta)-2Q_\lambda(\cos\theta))}{2(1+\lambda)(P_\lambda(\cos\xi)Q_{\lambda+1}(\cos\xi)-P_{\lambda+1}(\cos\xi)Q_\lambda(\cos\xi))}, & \theta > \xi \\[4mm] \dfrac{P_\lambda(\cos\theta)(\pi\cot(\pi\lambda)P_\lambda(\cos\xi)-2Q_\lambda(\cos\xi))}{2(1+\lambda)(P_\lambda(\cos\xi)Q_{\lambda+1}(\cos\xi)-P_{\lambda+1}(\cos\xi)Q_\lambda(\cos\xi))}, & \theta < \xi \end{cases}.
\tag{11}
$$

Here $P_\lambda$ and $Q_\lambda$ are Legendre functions of order $\lambda$, where

$$
\lambda = \frac{1}{2}\left(\sqrt{1-4\beta}-1\right).
\tag{12}
$$

This Green's function is continuous and bounded on the domain $\theta \in [0,\pi]$, and its derivative is also bounded at the boundaries $\theta = 0$ and $\theta = \pi$, for a given $\xi \in [0,\pi]$.

## 2.1 No partial ice cover

The BIM relies on initially positing an ansatz regarding the distribution of ice and water, and then the domain is partitioned into regions where the albedo function remains invariant with respect to temperature. Subsequently, the identity (10) is applied within these regions to obtain explicit expressions for the solution to (3). We start by analyzing solutions where the surface has no partial ice cover. This is the linear problem where the ice albedo feedback is inactive due to extreme temperatures. The step function albedo (2) leads to

$$
h(T,\theta) = \begin{cases} \eta s(\theta)(1-a_1)-\alpha, & T > -1 \\ \eta s(\theta)(1-a_2)-\alpha, & T < -1. \end{cases}
\tag{13}
$$

Note that $h$ is a function of non-dimensional temperature; hence, the critical temperature for the presence of surface ice is $T = -1$. For theses extreme cases, where $T > -1 \,\forall\, \theta \in [0,\pi]$ and $T < -1 \,\forall\, \theta \in [0,\pi]$, the surface is either 1) devoid of ice entirely or 2) entirely covered by ice. Consequently, the function $h$ is constant on the domain $[0,\pi]$ and we may apply the relation (10) on the full domain: Letting $\theta_1 \to 0^+$ and $\theta_2 \to \pi^-$ we get

$$
\begin{aligned}
T(\xi) = {} & \int_0^\pi d\theta\, \sin\theta\, K(\theta,\xi)h(T,\theta) + \lim_{\theta_2\to\pi^-} K(\theta_2,\xi)\sin\theta_2 \frac{\partial T}{\partial\theta}(\theta_2) \\
& - \lim_{\theta_2\to\pi^-} T(\theta_2)\sin\theta_2 \frac{\partial K}{\partial\theta}(\theta_2,\xi) - \lim_{\theta_1\to 0^+} K(\theta_1,\xi)\sin\theta_1 \frac{\partial T}{\partial\theta}(\theta_1) \\
& + \lim_{\theta_1\to 0^+} T(\theta_1)\sin\theta_1 \frac{\partial K}{\partial\theta}(\theta_1,\xi).
\end{aligned}
\tag{14}
$$

The Green's function, $K$, and its derivative are bounded at the boundary (see Appendix A). The gradient must vanish at the boundaries (North, 1975a) as we allow for no heat transport at the poles, leading to the boundary conditions,

$$
\lim_{\theta\to 0} \sin\theta\, \frac{\partial T}{\partial\theta}(\theta) = 0
\tag{15}
$$

and

$$\lim_{\theta \to \pi} \sin \theta \frac{\partial T}{\partial \theta}(\theta) = 0. \tag{16}$$

This ensures that Eq. (14) takes the simpler form

$$T(\xi) = \int_0^\pi d\theta \, \sin \theta \, K(\theta, \xi) h(T, \theta). \tag{17}$$

Using this, we may express the solution to Eq. (3) as

$$T(\xi) = \int_0^\pi d\theta \, \sin \theta \, K(\theta, \xi) h_1(\theta) \tag{18}$$

for case 1) and

$$T(\xi) = \int_0^\pi d\theta \, \sin \theta \, K(\theta, \xi) h_2(\theta) \tag{19}$$

for case 2), where $h_1 = \eta s(\theta)(1 - a_1) - \alpha$ and $h_2 = \eta s(\theta)(1 - a_2) - \alpha$.

## 2.2 Partial ice cover

For solutions to Eq. (1) where the zonal mean temperature profile is not strictly above or below the critical temperature, $T_s$, the surface will have a partial ice cover analogous to the Earth's current climate state. Critical latitudes, denoted as $\theta_{c_1}$ and $\theta_{c_2}$, mark the transitions between ice and water coverage. Assuming that $T$ remains continuous across the critical latitudes, the non-dimensional temperature at these latitudes must necessarily be $T(\theta_{c_1}) = T(\theta_{c_2}) = -1$. Given that the radiation distribution $s(\theta)$ prescribes an incoming radiation maximum at the equator, the ice cover must be centered at the poles (as illustrated in
Fig. 1, but without the continent), and it is sensible to partition the domain into the subdomains:

$$\theta \in (0, \theta_{c_1}), \tag{20}$$

$$\theta \in (\theta_{c_1}, \theta_{c_2}) \tag{21}$$

and

120 $$\theta \in (\theta_{c_2}, \pi), \tag{22}$$

such that

$$\mathcal{L}T = h_1. \tag{23}$$

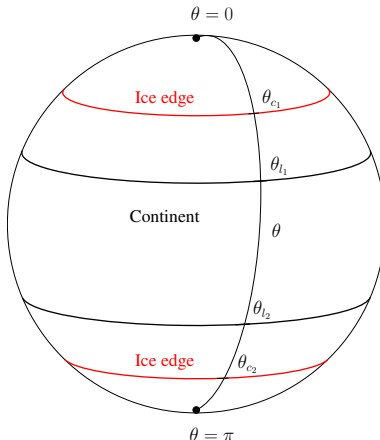

$\theta = 0$

Ice edge   $\theta_{c_1}$

$\theta_{l_1}$

Continent   $\theta$

$\theta_{l_2}$

Ice edge   $\theta_{c_2}$

$\theta = \pi$

**Figure 1.** Schematic of the domain $\theta \in [0, \pi]$ of Eq. (1) for a planet with a zonally symmetric continent and partial ice cover. The extent of the continent, i.e. $\theta_{l_1}$ and $\theta_{l_2}$, is determined by Eq. (34). The ice caps extend from $\theta = 0$ to the critical latitude $\theta = \theta_{c_1}$ and from $\theta = \theta_{c_2}$ to $\theta = \pi$.

in region (21) and

$$\mathcal{L}T = h_2 \tag{24}$$

in region (20) and (22).

The relation (10) is applied in these regions, and we get

$$T(\xi) = \int_0^{\theta_{c_1}} d\theta\, \sin\theta\, K(\theta,\xi) h_2(\theta) + K(\theta_{c_1},\xi)\sin(\theta_{c_1})\frac{\partial T}{\partial \theta}(\theta_{c_1}) + \sin(\theta_{c_1}) \lim_{\theta \to \theta_{c_1}^-} \frac{\partial K}{\partial \theta}(\theta,\xi), \tag{25}$$

$$\begin{aligned} T(\xi) = \int_{\theta_{c_1}}^{\theta_{c_2}} d\theta\, \sin\theta\, K(\theta,\xi) h_1(\theta) + K(\theta_{c_2},\xi)\sin(\theta_{c_2})\frac{\partial T}{\partial \theta}(\theta_{c_2}) + \sin(\theta_{c_2}) \lim_{\theta \to \theta_{c_2}^-} \frac{\partial K}{\partial \theta}(\theta,\xi) \\ - K(\theta_{c_1},\xi)\sin(\theta_{c_1})\frac{\partial T}{\partial \theta}(\theta_{c_1}) - \sin(\theta_{c_1}) \lim_{\theta \to \theta_{c_1}^+} \frac{\partial K}{\partial \theta}(\theta,\xi), \end{aligned} \tag{26}$$

and

$$T(\xi) = \int_{\theta_{c_2}}^{\pi} d\theta\, \sin\theta\, K(\theta,\xi) h_2(\theta) - K(\theta_{c_2},\xi)\sin(\theta_{c_2})\frac{\partial T}{\partial \theta}(\theta_{c_2}) - \sin(\theta_{c_2}) \lim_{\theta \to \theta_{c_2}^+} \frac{\partial K}{\partial \theta}(\theta,\xi) \tag{27}$$

in region (20), (21) and (22), respectively. The solution to Eq. (3) within the three subdomains may be expressed through relation (25)–(27) given the points $\theta_{c_1}$ and $\theta_{c_2}$, as well as the spatial derivative of $T$ at these points. These unknown boundary

values are determined by solving the following system of boundary integral equations, obtained by letting $\xi$ approach the boundaries of the three subdomains in Eq. (25)–(27):

$$
T(0) = \int_0^{\theta_{c_1}} d\theta \, \sin\theta \, K(\theta,0) h_2(\theta)
$$

$$
+ K(\theta_{c_1},0)\sin(\theta_{c_1})\frac{\partial T}{\partial \theta}(\theta_{c_1}) + \sin(\theta_{c_1}) \lim_{\xi \to 0^+} \lim_{\theta \to \theta_{c_1}^-} \frac{\partial K}{\partial \theta}(\theta,\xi)
$$

(28)

$$
-1 = \int_0^{\theta_{c_1}} d\theta \, \sin\theta \, K(\theta,\theta_{c_1}) h_2(\theta)
$$

$$
+ K(\theta_{c_1},\theta_{c_1})\sin(\theta_{c_1})\frac{\partial T}{\partial \theta}(\theta_{c_1}) + \sin(\theta_{c_1}) \lim_{\xi \to \theta_{c_1}^-} \lim_{\theta \to \theta_{c_1}^-} \frac{\partial K}{\partial \theta}(\theta,\xi)
$$

(29)

$$
-1 = \int_{\theta_{c_1}}^{\theta_{c_2}} d\theta \, \sin\theta \, K(\theta,\theta_{c_1}) h_1(\theta)
$$

$$
+ K(\theta_{c_2},\theta_{c_1})\sin(\theta_{c_2})\frac{\partial T}{\partial \theta}(\theta_{c_2}) + \sin(\theta_{c_2}) \lim_{\xi \to \theta_{c_1}^+} \lim_{\theta \to \theta_{c_2}^-} \frac{\partial K}{\partial \theta}(\theta,\xi)
$$

$$
- K(\theta_{c_1},\theta_{c_1})\sin(\theta_{c_1})\frac{\partial T}{\partial \theta}(\theta_{c_1}) - \sin(\theta_{c_1}) \lim_{\xi \to \theta_{c_1}^+} \lim_{\theta \to \theta_{c_1}^+} \frac{\partial K}{\partial \theta}(\theta,\xi)
$$

(30)

$$
-1 = \int_{\theta_{c_1}}^{\theta_{c_2}} d\theta \, \sin\theta \, K(\theta,\theta_{c_2}) h_1(\theta)
$$

$$
+ K(\theta_{c_2},\theta_{c_2})\sin(\theta_{c_2})\frac{\partial T}{\partial \theta}(\theta_{c_2}) + \sin(\theta_{c_2}) \lim_{\xi \to \theta_{c_2}^-} \lim_{\theta \to \theta_{c_2}^-} \frac{\partial K}{\partial \theta}(\theta,\xi)
$$

$$
- K(\theta_{c_1},\theta_{c_2})\sin(\theta_{c_1})\frac{\partial T}{\partial \theta}(\theta_{c_1}) - \sin(\theta_{c_1}) \lim_{\xi \to \theta_{c_2}^-} \lim_{\theta \to \theta_{c_1}^+} \frac{\partial K}{\partial \theta}(\theta,\xi)
$$

(31)

$$
-1 = \int_{\theta_{c_2}}^{\pi} d\theta \, \sin\theta \, K(\theta,\theta_{c_2}) h_2(\theta)
$$

$$
- K(\theta_{c_2},\theta_{c_2})\sin(\theta_{c_2})\frac{\partial T}{\partial \theta}(\theta_{c_2}) - \sin(\theta_{c_2}) \lim_{\xi \to \theta_{c_2}^+} \lim_{\theta \to \theta_{c_2}^+} \frac{\partial K}{\partial \theta}(\theta,\xi)
$$

(32)

$$
T(\pi) = \int_{\theta_{c_2}}^{\pi} d\theta \, \sin\theta \, K(\theta,\pi) h_2(\theta)
$$

$$
- K(\theta_{c_2},\pi)\sin(\theta_{c_2})\frac{\partial T}{\partial \theta}(\theta_{c_2}) - \sin(\theta_{c_2}) \lim_{\xi \to \pi^-} \lim_{\theta \to \theta_{c_2}^+} \frac{\partial K}{\partial \theta}(\theta,\xi)
$$

(33)

The system of equations (28)–(33) can be solved for $T(0)$, $T(\pi)$, $\frac{\partial T}{\partial \theta}(\theta_{c_1})$, $\frac{\partial T}{\partial \theta}(\theta_{c_2})$, $\theta_{c_1}$ and $\theta_{c_2}$ through a combination of both analytical and numerical methods. Finding $\theta_{c_1}$ and $\theta_{c_2}$ ultimately requires a root search, as the integrals cannot be evaluated analytically. However, these values may be approximated to a high precision. For the root search, we used a Broyden's method, as computing the Jacobian is expensive for solutions with multiple critical latitudes. Note that the system of equations (28)–(33) can have more than one solution, indicating multiple equilibria.

## 2.3 With a continent

The method presented may be extended to include one or more zonally symmetric continents. The following analysis includes one such continent with a meridional extent of $l = \frac{\pi}{4}$, stretching from latitude $\theta_{l_1}$ to latitude $\theta_{l_2}$. Figure 1 illustrates a planet with a continent of this kind. The analysis was repeated three times for three different continent configurations,

$$
\begin{aligned}
\theta_{l_1} &= \frac{\pi}{2} - \frac{l}{2} - \varepsilon \\
\theta_{l_2} &= \frac{\pi}{2} + \frac{l}{2} - \varepsilon,
\end{aligned}
\tag{34}
$$

where $\varepsilon = 0$, $\varepsilon = 0.1$ and $\varepsilon = 0.5$. Parameter values on the continent may be altered to distinguish land from ocean and better capture the thermal response of the lithosphere. Here, the heat capacity $C$, the critical temperature for ice formation $T_s$ and the albedo $a(T)$ were altered on the part of the domain corresponding to the continent. This has the effect of changing the ice dynamics and subsequently the ice-albedo feedback, on the continent. The extent of the continent $l$ is kept constant, and no ice-ocean-land feedback is included in the model. Mathematical details on the application of the presented method to model configurations with a continent are omitted for brevity. Instead, we provide some results of our analysis. Interested readers are referred to Samuelsberg and Jakobsen (2023) for a full derivation of these solutions. The multiple branch structure of the model is displayed in Fig. 2 through bifurcation diagrams of the system with a continent configuration as in Eq. (34), where $\varepsilon = 0$, $\varepsilon = 0.1$ and $\varepsilon = 0.5$. The control parameter is the scaled TSI $Q$. Stability properties of the stationary solutions are assessed using the numerical perturbation scheme outlined in Appendix B.

## 3 Discussion

We have applied the BIM to solve the stationary form of a North-type EBM with a zonally symmetric continent in three different configurations. The introduction of a continent resulted in the emergence of new equilibrium states. EMBs have a rich multiple solution structure (North, 1990). In zero-dimensional models that incorporate the ice albedo feedback, three solutions exist, and as one extends to one-dimensional models the multiple branch structure becomes more complicated. North showed that there exist up to five solutions for a range of TSI values in the globally averaged model, one of which is the famously unstable small ice cap solution (North, 1984). From Fig. 2 (a), it is evident that up to seven equilibria may exist for a range of $Q$ values in model configurations with a continent. The continent is initially placed in a North-South symmetrical configuration to investigate how the system is affected by symmetry. The meridional symmetry is subsequently violated by increasing $\varepsilon$. Although the number of equilibria for any given $Q$ is at most seven for $\varepsilon = 0$ and $\varepsilon = 0.1$, the range over which seven equilibria can exist is reduced

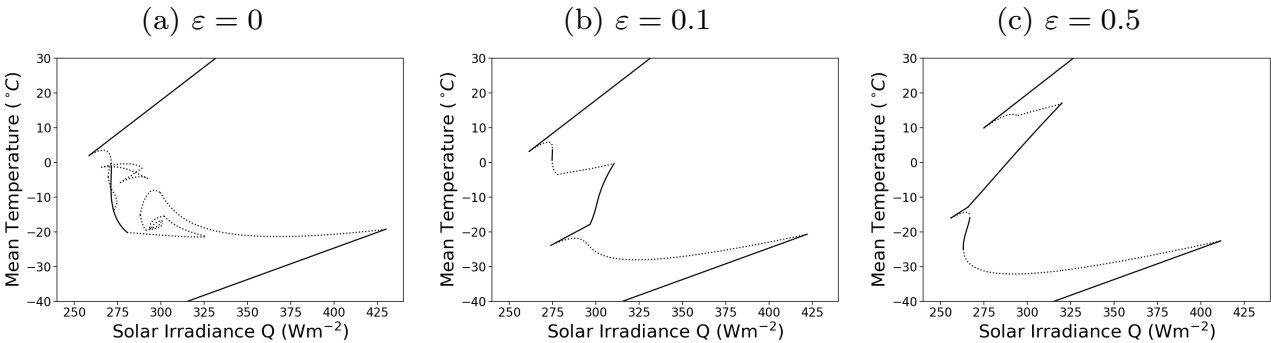

**Figure 2.** Bifurcation diagrams. Annual mean equilibrium surface temperatures plotted against the control parameter $Q$, for model configurations with a continent as in (34), where $\varepsilon = 0$, $\varepsilon = 0.1$ and $\varepsilon = 0.5$. Model parameters are those in Table 1. Solid lines indicate stable solutions and dotted lines are unstable solutions. The upper, stable branches in the bifurcation diagrams contain ice-free solutions and the lower, stable branches contain solutions with a full ice cover (Snowball Earth states). Intermediate branches contain solutions with a partial ice cover.

as $\varepsilon$ is increased and has disappeared for $\varepsilon = 0.5$. Prescribing the system inherently symmetrical boundary conditions, i.e., $\varepsilon = 0$, evidently introduces a very fine dynamic as seen in the looping branches in the bifurcation diagram Fig. 2 (a), which disappears for non-zero $\varepsilon$. Furthermore, equilibria with a more complicated ice distribution, characterized by more than two critical latitudes, only appear for $\varepsilon = 0$. Despite this, the range of TSI over which stable partial-ice-cover solutions can exist, that is, the intermediate branches in Fig. 2, is much shorter for $\varepsilon = 0$. The range of stable, intermediate branches is longest for $\varepsilon = 0.5$. Additionally, for $\varepsilon = 0.5$, there is a large range of TSI values over which bi-stability occurs between the ice-free solution and the partial-ice-cover solution, as seen in Fig. 2 (c). These observations are notable because, although it is well established that the climate has changed throughout geological history, the role of land-sea distribution is not fully understood (Fluteau, 2003).

The robustness of the method was tested by allowing for parameter discontinuities at the land-sea boundaries. Mengel et al. (1988) and North and Kim (2017) have discussed the application of Fourier-Legendre series in EBMs with parameter discontinuities at the continent edges: A discontinuity in the albedo and heat capacity parameter $C$ causes a solution expressed through Legendre modes to converge slowly. Presumably, similar discontinuities in other parameters will have the same effect. Moreover, North (1975a) discussed the potential for changing parameter values on finite zonal strips using spectral methods, but did not address how this may result in several ice edges. The dynamics of EBMs are highly sensitive to model parameters (Soldatenko and Colman, 2019), and the introduction of a continent to the model can result in some unusual ice distributions. Figure 3 shows an equilibrium solution with six critical latitudes and two ice belts on the continent. Studying several critical latitudes is a natural extension of the BIM and follows the same general procedure. A similar ice belt has been observed by changing the obliquity of the model using the spectral method (Rose et al., 2017). For high obliquities, the traditional ice distribution of the two-critical-latitude solution is inverted, allowing for a similar analysis to the classical partial-ice-

cover states, without additional ice edges. The ice belts overlaying the continent form a striped pattern, a phenomenon also observed in the related one-dimensional Daisyworld model with diffusion (Adams et al., 2003; Alberti et al., 2015). Adams et al. (2003) found that the daisies never coexisted; instead the equilibrium solution is characterized by zonal bands of single-species colonies reminiscent of Fig. 3. A similar striped pattern of daisy coverage was reported by Alberti et al. (2015) in a one-dimensional Daisyworld model with diffusion and a greenhouse effect.

Although the BIM is an effective approach for solving North-type EBMs, it has certain limitations. A step function albedo is used here, and the method is easily extended to any latitude dependence for the albedo. However, this method, like the spectral method, is limited to a step function-like temperature response at the ice edge. An arbitrary temperature-dependent albedo on either side of the ice edge renders the energy balance equation unsolvable through the presented method. An additional drawback of the presented method is that a root search is required to find the critical latitudes, $\theta_{c_i}$. For solutions with a low number of critical latitudes, this poses no significant challenge. However, as the number of critical latitudes increases, so does the complexity of the root search and the necessity for a good starting point in the iteration. For solutions located in branches connected to the ice-free branch and the Snowball Earth branch in the bifurcation diagram, this is generally not a problem. Since these extreme solutions always exist within certain parameter regimes, we identify the bifurcation points and update the ansatz accordingly, repeating this process until the upper and lower branches connect. This approach ensures sufficient initial points for the root searches, allowing all co-existing states in connected branches to be identified. Additionally, for solutions with one, two, or three critical latitudes, a graphical solution to the system of boundary integral equations (Eq. (28)–(33)) is possible, ensuring that any isolated branches are also detected. However, to detect isolated branches with multiple critical latitudes requires a more numerically expensive approach. Furthermore, the method can become tedious for complicated geographies and a high number of critical latitudes.

The BIM has a wide range of potential applications. While the model studied here only includes an ice-albedo feedback mechanism, the BIM can be generalized to solve one-dimensional EBMs with additional albedo feedbacks. A vegetation feedback has previously been introduced within an EBM framework for zero-dimensional models (Rombouts and Ghil, 2015; Alberti et al., 2018) and related one-dimensional models (Wood et al., 2008; Adams et al., 2003; Alberti et al., 2015). The method we have presented is applicable to North-type EBMs where the model is piecewise linear on subdomains, specifically if the explicit temperature dependence in the integrand of Eq. (10) can be eliminated through an appropriate partitioning of the domain. For instance, a vegetation feedback mechanism can be implemented by adding vegetation on the continent for temperatures within a certain growth regime, where the vegetation is modeled by altering the albedo. The BIM can be directly applied to North-type models with such simplified vegetation responses. Furthermore, including additional spatial dependence in the albedo represents another natural application of the BIM. For example, in studies involving North-type EBMs related to Snowball Earth events, alternative albedo parameterizations (Abbot et al., 2011) or spatial dependence for other model parameters can be easily implemented with the BIM. The ability to effectively handle spatially dependent model parameters also makes the BIM an appropriate tool for studying fragmented tipping in North-type EBMs (Bastiaansen et al., 2022).

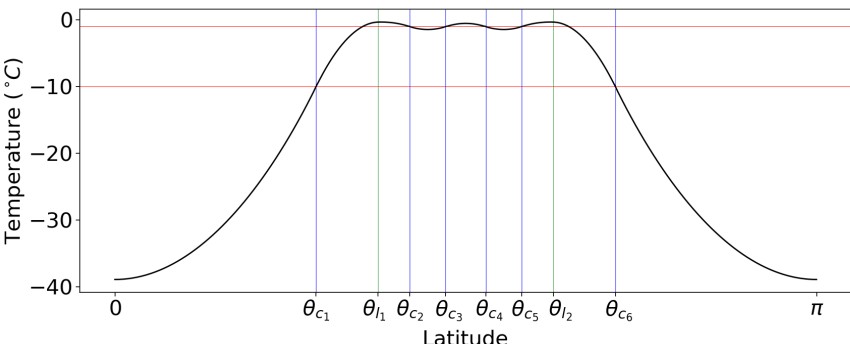

**Figure 3.** An equilibrium solution (unstable) to Eq. (1) with a continent as in (34), where $\varepsilon = 0$, $Q = 294\,\mathrm{Wm^{-2}}$ and other parameters are those in Table 1. Green vertical lines mark the continent borders, and blue vertical lines mark critical latitudes. The red horizontal lines mark the critical temperatures for ice formation.

## 4 Conclusions

In this paper, we have presented an analytical method for solving North-type EBMs. Solutions are expressed through explicit expressions, readily obtainable from quadrature methods. The presented method has some notable advantages compared to other analytical methods, e.g., (North, 1975a), for solving energy balance equations of this kind. It does not rely on truncating series expansions. Furthermore, the method remains straightforward, and computationally speaking, very fast, even for problems with partial land-sea geographies and parameter discontinuities at the boundaries separating land and sea. In addition, the BIM offers a formulaic framework for handling equilibrium solutions with several critical latitudes $\theta_{c_i}$ and unconventional ice distributions.

The development of new analytical methods of studying EBMs is motivated by the recognition of EBMs as useful tools for researchers in a variety of fields. The inherent simplicity of EBMs, characterized by few parameters, renders them particularly suitable for certain aspects of paleoclimatology (North and Kim, 2017; Abbot et al., 2011; Widiasih et al., 2024) and planetary science (Rose et al., 2017), where poorly constrained parameters and a diverse set of planetary conditions are frequently encountered. The BIM represents a systematic approach to solving North-type EBMs and excels under unconventional models configurations, particularly where emerging solutions describe climate states markedly different from the prevailing state of Earth. Analytical investigations of conceptual models continue to provide a valuable testing ground for ideas in climate science and insights into the complex dynamics involved as one ascends the climate model hierarchy.

*Code availability.* The codes for solving the stationary form of Eq. (1) and for producing Fig. 2 and Fig. 3 in this manuscript are available on Zenodo (https://doi.org/10.5281/zenodo.11083624, Samuelsberg & Jakobsen, 2024).

**Table 1.** Model parameters used in the presented work. $A$, $B$, $C$, $C_{\text{land}}$ and $T_s$ are taken from North et al. (1981). $D$ is taken from Kaper and Engler (2013) and $s(\theta)$ is taken from McGehee and Lehman (2012).

| Parameter | Value |
|-----------|-------|
| $s(\theta)$ | $s_0 + s_1 \cos^2(\theta - \frac{\pi}{2})$ |
| $s_0$ | 0.523 |
| $s_1$ | 0.716 |
| $A$ | $203\,\text{W}\,\text{m}^{-2}$ |
| $B$ | $2.09\,\text{W}\,\text{m}^{-2}(^{\circ}\text{C})^{-1}$ |
| $D$ | $0.208 \cdot B$ |
| $C$ | $4.7\,B\,t_0$ |
| $C_{\text{land}}$ | $0.16\,B\,t_0$ |
| $t_0$ | 1 year |
| $T_s$ | $10\,^{\circ}\text{C}$ |
| $T_{s,\text{land}}$ | $1\,^{\circ}\text{C}$ |
| $a_1$ | 0.06 |
| $a_2$ | 0.6 |
| $a_{1,\text{land}}$ | 0.3 |
| $a_{2,\text{land}}$ | 0.6 |

## Appendix A: Finding a Green's function

In this section, we find a Green's function, $K(\theta, \xi)$, for the operator (4). A Green's function is any solution to the Eq. (8). The two defining properties of the Dirac-delta function are:

1. for any surface of interest, $S$, we must have

$$\int_S dA\, \delta_{\boldsymbol{\xi}} = 1, \tag{A1}$$

2. for any function, $f(\boldsymbol{x})$, defined on $S$ we must have

$$\int_S dA\, \delta_{\boldsymbol{\xi}} f = f(\boldsymbol{\xi}). \tag{A2}$$

For the line $\theta \in [0, \pi]$ along the surface of a sphere with radius $R$, it can be shown that

$$\delta_{\xi}(\theta) = \frac{\delta(\theta - \xi)}{2\pi R^2 \sin\theta}, \tag{A3}$$

where $\delta(\theta - \xi)$ is the usual delayed Dirac-delta function on the line, will ensure that Eq. (A1) and Eq. (A2) are satisfied. It is convenient to scale the Green's function we are seeking by a factor such that the right-hand side of Eq. (A3) becomes unity

and Eq. (8) becomes

$$\mathcal{L}K = \frac{\delta(\theta - \xi)}{\sin\theta}.$$

(A4)

We will demand that the Green's function is continuous across $\theta = \xi$, therefore we must have

$$\lim_{\theta \to \xi^+} K(\theta, \xi) = \lim_{\theta \to \xi^-} K(\theta, \xi).$$

(A5)

Integrating Eq. (A4) over a small interval centered on $\theta = \xi$ we get

$$\int_{\xi - \varepsilon}^{\xi + \varepsilon} d\theta \, \sin\theta \left\{ -\frac{1}{\sin\theta} \frac{\partial}{\partial\theta} \left( \sin\theta \frac{\partial K}{\partial\theta} \right) + \beta K \right\} = \int_{\xi - \varepsilon}^{\xi + \varepsilon} d\theta \, \sin\theta \frac{\delta(\theta - \xi)}{\sin\theta}$$

(A6)

$$-\int_{\xi - \varepsilon}^{\xi + \varepsilon} d\theta \, \frac{\partial}{\partial\theta} \left( \sin\theta \frac{\partial}{\partial\theta} K(\theta, \xi) \right) + \beta \int_{\xi - \varepsilon}^{\xi + \varepsilon} d\theta \, \sin\theta \, K(\theta, \xi) = 1$$

(A7)

$$-\sin\theta \frac{\partial}{\partial\theta} K(\theta, \xi) \Big|_{\xi - \varepsilon}^{\xi + \varepsilon} + \beta \int_{\xi - \varepsilon}^{\xi + \varepsilon} d\theta \, \sin\theta \, K(\theta, \xi) = 1.$$

(A8)

Letting $\varepsilon \to 0$ we must have

$$\lim_{\theta \to \xi^+} \frac{\partial}{\partial\theta} K(\theta, \xi) - \lim_{\theta \to \xi^-} \frac{\partial}{\partial\theta} K(\theta, \xi) = -\frac{1}{\sin\xi}.$$

(A9)

At $\theta \neq \xi$ we evidently have

$$\mathcal{L}K(\theta, \xi) = 0.$$

(A10)

We can therefore conclude that $K(\theta, \xi)$ must satisfy the necessary conditions (A5), (A9) and (A10). In order to find a Green's function that solves Eq. (A4) we need to find a basis of solutions for an equation on the form

$$-\frac{1}{\sin\theta} \frac{\partial}{\partial\theta} \left( \sin\theta \frac{\partial}{\partial\theta} y(\theta) \right) + \beta y(\theta) = 0.$$

(A11)

Introducing a change of variables, $x = \cos\theta$, and a function, $u$, such that $u(\cos\theta) = y(\theta)$ it can be shown that Eq. (A11) can be written on the form

$$(1 - x^2) \frac{\partial^2}{\partial x^2} u(x) + 2x \frac{\partial}{\partial x} u(x) - \beta u(x) = 0.$$

(A12)

Let $\lambda$ be a number such that $-\beta = \lambda(\lambda + 1)$. We may now write Eq. (A12) as a Legendre equation,

$$(1 - x^2) \frac{\partial^2}{\partial x^2} u(x) + 2x \frac{\partial}{\partial x} u(x) + \lambda(\lambda + 1) u(x) = 0,$$

(A13)

which for some arbitrary real or complex value $\lambda$, will have the known basis of solutions $\{P_\lambda, Q_\lambda\}$. We can therefore use the basis $\{P_\lambda(\cos\theta), Q_\lambda(\cos\theta)\}$, where $\lambda = \frac{1}{2}\left(\sqrt{1 - 4\beta} - 1\right)$, to construct the general solution to Eq. (A10),

$$K(\theta, \xi) = \begin{cases} a(\xi) P_\lambda(\cos\theta) + b(\xi) Q_\lambda(\cos\theta), & \theta > \xi \\ c(\xi) P_\lambda(\cos\theta) + d(\xi) Q_\lambda(\cos\theta), & \theta < \xi \end{cases}.$$

(A14)

The coefficients $a(\xi), b(\xi), c(\xi)$ and $d(\xi)$ can be determined through the conditions (A5) and (A9). Any choice of these coefficients satisfying Eq. (A5) and Eq. (A9) will give a Green's function for the operator (4). However, it makes sense for us to seek a Green's function that is non-singular in the domain $\theta \in [0, \pi]$: We want to to develop a set of conditions on the coefficients $a(\xi), b(\xi), c(\xi)$ and $d(\xi)$ to ensure that the Green's function (A14) is non-singular when $\theta \to 0$ and $\theta \to \pi$. Let

$$
\begin{aligned}
K_+(\theta, \xi) &= a(\xi) P_\lambda(\cos\theta) + b(\xi) Q_\lambda(\cos\theta) \\
K_-(\theta, \xi) &= c(\xi) P_\lambda(\cos\theta) + d(\xi) Q_\lambda(\cos\theta)
\end{aligned}
\tag{A15}
$$

such that

$$
K(\theta, \xi) = \begin{cases} K_+(\theta, \xi), & \theta > \xi \\ K_-(\theta, \xi), & \theta < \xi \end{cases}.
\tag{A16}
$$

Using a computer algebra system, we find a series expansion of $K_+$ around $\theta = \pi$, and recognize that there is a term in this expansion containing $\log(\pi - \theta)$ with a coefficient $c_{+0}(a(\xi), b(\xi))$. We want to ensure that $K_+$ is non-singular at $\theta = \pi$ and therefore demand that

$$
c_{+0}(a(\xi), b(\xi)) = 0 \,\forall\, \xi \in [0, \pi].
\tag{A17}
$$

Similarly, we find a series expansion of $K_-$ around $\theta = 0$. In this expansion there is a term containing $\log(\theta)$ with a coefficient $c_{-0}(c(\xi), d(\xi))$, and we demand that

$$
c_{-0}(c(\xi), d(\xi)) = 0 \,\forall\, \xi \in [0, \pi].
\tag{A18}
$$

Solving the system of equations (A5), (A9), (A17) and (A18) for $a(\xi), b(\xi), c(\xi)$ and $d(\xi)$, we find the following Green's function;

$$
K(\theta, \xi) = \begin{cases} \dfrac{P_\lambda(\cos\xi)(\pi \cot(\pi\lambda) P_\lambda(\cos\theta) - 2Q_\lambda(\cos\theta))}{2(1+\lambda)(P_\lambda(\cos\xi) Q_{\lambda+1}(\cos\xi) - P_{\lambda+1}(\cos\xi) Q_\lambda(\cos\xi))}, & \theta > \xi \\[4mm] \dfrac{P_\lambda(\cos\theta)(\pi \cot(\pi\lambda) P_\lambda(\cos\xi) - 2Q_\lambda(\cos\xi))}{2(1+\lambda)(P_\lambda(\cos\xi) Q_{\lambda+1}(\cos\xi) - P_{\lambda+1}(\cos\xi) Q_\lambda(\cos\xi))}, & \theta < \xi \end{cases}.
\tag{A19}
$$

The Green's function (A19) is non-singular and bounded at $\theta = 0$ and $\theta = \pi$. The derivative of the Green's function (A19) is also bounded at the boundary and tends to zero $\forall\, \xi \in [0, \pi]$.

## Appendix B: Stability analysis

In this section, we are going to test the stability of the stationary solutions found using the BIM. Applying the notation from section 2 we may write the time dependent form of Eq. (1) as

$$
\gamma \partial_t T + \mathcal{L} T = h(T, \theta),
\tag{B1}
$$

where $\gamma = \frac{C}{t_0 D}$. Stationary solutions are denoted $T_0 = T(\theta, t = 0)$, such that $\mathcal{L} T_0 = h$. We wish to investigate whether a small perturbation, $\delta$, away from the equilibrium, $T_0$, will grow in time. The perturbed solution, $T(\theta, t) = T_0(\theta) + \delta(\theta, t)$, inserted into Eq. (B1) yields

$$\gamma \partial_t \delta + \mathcal{L} T_0 + \mathcal{L} \delta = h(T_0 + \delta, \theta). \tag{B2}$$

A first order expansion around $(T_0, \theta)$ of the right-hand side may be expressed as

$$h(T_0 + \delta, \theta) \approx h(T_0, \theta) + h_T(T_0, \theta)\delta. \tag{B3}$$

Here we let

$$a(T) = a_1 + \frac{a_2 - a_1}{2}(1 + \tanh(-\sigma(T + 1))), \tag{B4}$$

where the slope parameter $\sigma = 50$, be a smooth function replicating the behavior of the step function albedo such that the derivative $h_T(T_0, \theta) = \frac{\partial h}{\partial T}(T_0, \theta)$ may be found analytically for a given $T_0$. By substituting Eq. (B3) into Eq. (B2) we get

$$\partial_t \delta = \mathcal{H} \delta, \tag{B5}$$

where

$$\mathcal{H}(\cdot) = \frac{1}{\gamma}\left[ h_T(T_0, \theta)(\cdot) - \mathcal{L}(\cdot) \right]. \tag{B6}$$

Suppose that the perturbation $\delta$ has the form

$$\delta(\theta, t) = e^{\lambda t} \delta_0(\theta). \tag{B7}$$

This turns Eq. (B5) into the following eigenvalue problem

$$\lambda \delta_0 = \mathcal{H} \zeta_0. \tag{B8}$$

Real and positive $\lambda$ will evidently cause the perturbation to grow exponentially, resulting in an unstable the stationary solution $T_0$. The eigenvalues were subsequently approximated using a numerical scheme that solves the associated eigenvalue problem

$$\lambda \boldsymbol{\delta_0} = H \boldsymbol{\delta_0}, \tag{B9}$$

where the set of linear equations

$$\left\{ \lambda \delta_0^i = \frac{1}{\gamma}\left[ h_T(T_0^i, \theta_i)\delta_0^i - \hat{\mathcal{L}}\delta_0^i \right] \right\}_{i=0}^{N} \tag{B10}$$

gives rise to the coefficient matrix $H$. Here $T_0^i$ is the stationary solution evaluated on a uniform spatial grid and $\hat{\mathcal{L}}(\cdot)$ is a finite difference approximation of the differential operator $\mathcal{L}$. It can be shown that a second-order centered difference approximation for a smooth function $\delta_0$, evaluated on a uniform grid $\theta_i$, where $\delta_0^i = \delta_0(\theta_i)$, is

$$\hat{\mathcal{L}}\delta_0^i = \beta \delta_0^i - \frac{2(\delta_0^{i-1} - 2\delta_0^i + \delta_0^{i+1}) + (\delta_0^{i-1} - 4\delta_0^i + 3\delta_0^{i+1})d\theta \cot \theta_i}{2d\theta^2}. \tag{B11}$$

At either end of the grid a forward and backward approximation is needed. These are

$$\hat{\mathcal{L}}_f \delta_0^0 = \beta \delta_0^0 + \frac{-2(\delta_0^0 - 2\delta_0^1 + \delta_0^2) + (5\delta_0^0 - 8\delta_0^1 + 3\delta_0^2)d\theta \cot \theta_0}{2d\theta^2} \tag{B12}$$

and

$$\hat{\mathcal{L}}_b \delta_0^N = \beta \delta_0^N + \frac{-2(\delta_0^{N-2} - 2\delta_0^{N-1} + \delta_0^N) + (\delta_0^{N-2} - \delta_0^N)d\theta \cot \theta_N}{2d\theta^2}. \tag{B13}$$

for the forward and backward approximation, respectively. Furthermore, as $t$ grows the perturbation (B7) must adhere to the same constraints as the solution, i.e. the boundary conditions (15) and (16). A discrete formulation of these is $\delta_0^0 = \delta_0^2$ and $\delta_0^N = \delta_0^{N-2}$, which we ensure is enforced. We build the matrix $H$ and examine the associated eigenvalues for a large number of points in the bifurcation diagram. Stability properties are subsequently inferred from the ensemble of stationary solutions within the same branch. The presented stability analysis agrees with the slope-stability theorem put forth by Cahalan and North

(1979).

*Author contributions.* PKJ conceived of the study. AS performed the calculations and wrote the manuscript.

*Competing interests.* The authors have no competing interests.

*Acknowledgements.* We would like to thank Francesco Berrilli and five anonymous reviewers for their valuable feedback and insightful comments, which have significantly contributed to improving the quality of this manuscript. This work was supported by the UiT Aurora Centre

Program, UiT The Arctic University of Norway (2024), and the Research Council of Norway (project number 314570). AS acknowledges the assistance of an AI-based tool, ChatGPT, for the refinement of the language in parts of this manuscript.

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
