# Peer review of "Solving a North-type energy balance model using boundary integral methods"

_Nonlinear Processes in Geophysics, 2024_

## Author Response (AR1)

**Author's response**

**Tracked changes**

Most changes to the manuscript are highlighted in blue, while text that has been removed is shown with a red strikethrough. Any changes not highlighted in blue are detailed below:

- The revised manuscript includes a more thorough discussion of the results. Therefore, the revision includes a "Discussion" section and parts of the "Conclusions" are moved to the "Discussion".

- All equations are now numbered.

- Additional references: Adams et al. 2003; Fluteau 2003; Hsiao and Wendland 2008; Morino and Piva 2012; E. R. Widiasih 2013; Walsh and E. Widiasih 2014; Alberti et al. 2015; Rombouts and Ghil 2015; Tommaso Alberti et al. 2018; E. R. Widiasih, Keane, and Stuecker 2024

- The labels on the x-axes of Fig. 2 have been changed from "Solar Constant Q" to "Solar Irradiance Q".

- Figure 3 has been improved by increasing the size of the tick marks on the x-axis.

**Reply to Reviewer 1**

**Answer:** We would like to express our gratitude to the Reviewer for their positive evaluation of the manuscript and valuable suggestions for its improvement. We have carefully considered all of the comments and have addressed them in the revised manuscript, as detailed in the responses below.

The manuscript introduces the boundary integral method (BIM) to find equilibrium solutions, and their corresponding stability properties, in North-type energy balance models (EBMs). I fully support the use of EBMs to understand and characterize the main features of climate states under different conditions. The manuscript is well written and it is surely appropriate for NPG. I have some moderate suggestions to compare the results of this paper with previously obtained ones in a similar framework.

**Main comments**

1. An interesting result is the "striped" pattern obtained in Figure 3. This pattern has been also reported in 1D EBMs including vegetation (e.g., Nevison et al., 1999; Adams et al., 2003; Alberti et al., 2015). I would suggest the authors to compare their results with those papers in terms of active feedbacks, especially those related to the ice-albedo feedback.
   **Answer:** The reviewer is correct that we missed some studies reporting similar striped patterns in 1D EBMs. We have compared our results to the suggested literature in the revised manuscript.

2. The authors consider a step-wise function for the albedo with a latitude dependence. However, they do not include in the feedback the extension of the continent (e.g., Wood et al., 2008; Rombouts and Ghil, 2015; Alberti et al., 2018). I would recommend to comment on this point and how the BIM can be generalized towards including additional contributions in the ice-albedo feedback.
   **Answer:** We thank the Reviewer for pointing this out. The BIM, as presented in the manuscript, can be generalized towards including additional albedo feedbacks, e.g. simplified vegetation response to temperature. This is discussed in the revision.

3. I found the results on the larger number of equilibria very interesting. I would suggest to include more discussion on the bifurcation diagrams reported in Figure 2. What about inspecting how the extension of the stable and unstable regimes depend on $\varepsilon$? This would suggest and provide more information on the residence time in a specific state for the model that can be useful for a broader discussion on the role of ice-ocean-land distribution on the climate states.
   **Answer:** We agree that the bifurcation diagrams in Fig. 2 can be further discussed. A more thorough discussion of these results is included in the revision, see the Discussion. Specifically, we have discussed how the parameter $\varepsilon$ affects the number of equilibria and their stability, as well as the range over which stable equilibria with a partial ice cover exist.

**Minor comments**

- Line 38: the values are slightly different from those usually reported in literature (Budyko, 1969; Rombouts and Ghil, 2015). Are these values corrected for current levels of greenhouse effect?
  **Answer:** The parameter values A and B are from North et al., 1981 and are therefore not corrected for current levels of greenhouse effect.

- Line 60: is in the definition of $\beta$ a missing Ts or Eq. (3) is just written in dimensionless temperature $(T \to T/Ts)$? Please clarify.
  **Answer:** Yes, Eq. (3) is written in terms of non-dimensional temperatures where the substitution $T_{dimensional} \to Ts T_{non-dimensional}$ has been applied. This is clarified in the revised manuscript where the non-dimensional temperature is initially given a tilde notation to distinguish it from the dimensional temperature. This notation is subsequently omitted for readability and a comment is added noting this.

- Eq. (8): please clarify that this is valid in a linear approximation around an equilibrium solution.
  **Answer:** The formulation prior to Eq. (8) was unclear. Eq. (8) is an identity the equilibrium solution $T$ must satisfy. This will be clarified in the revised manuscript.

- Eqs. (11)-(12): should these conditions valid for each selection of the ice-albedo feedback? Are there for boundary conditions at the poles, right?
  **Answer:** The no-flux boundary conditions at the poles Eqs. (11)-(12) are valid for bounded $h(T, \theta)$ and a bounded solution $T$. Thus they are valid for every selection of the ice-albedo feedback.

- Line 144: why choosing $l = \pi/4$? Please also clarify that in the ice-albedo feedback there is no ice-ocean-land feedback.
  **Answer:** The main point of adding a continent is to test the method presented. The extent of the continent $l = \pi/4$ was chosen arbitrarily. The Reviewer is correct that no ice-ocean-land feedback is included in the model and a comment has been added clarifying this.

- Figure 3: I would recommend to increase the quality of the figure since it seems that $\theta$ labels are not fully resolved.
  **Answer:** Done as suggested.

**Suggested references**

- Adams, B., et al. (2003) J. Theor. Biol., 223, 505.

- Adams, B., and Carr, J. (2003) Nonlinearity, 16, 1339.

- Alberti, T., et al. (2015) Phys. Rev. E, 92, 052717.

- Alberti, T., et al. (2017) ApJ, 844, 19.

- Alberti, T., et al. (2018) J. Phys. Comm., 2, 065018.

- Nevison, C., et al. (1999) Tellus B, 53, 288.

- Rombouts, J., and Ghil, M. (2015) NPG, 22, 275.

- Wood, A.J., et al. (2008) Rev. Geophys., 46, RG1001.

**Reply to Reviewer 2**

**Answer:** We sincerely thank the Reviewer for their constructive feedback and positive appraisal of our manuscript. We have carefully considered all of the comments and have addressed them in the revised version of the manuscript, as detailed in the responses below.

**General**

In the present manuscript the authors apply the boundary integral method (BIM) to find equilibrium solutions of a North-type one dimensional energy balance model. The proposed method is described in detail and applied using some case studies concerning the effect of a continent. The study demonstrates that BIM is a valuable method and may be applicable to conceptual models of similar type, which can still contribute significantly to our process understanding. The results of the showcase experiments indicate the existence a variety of equilibrium solutions depending on the location of the continent.

The manuscript is well written. The methodology is sound and sufficiently explained together with an appropriate application. Thus, in general, I can recommend publication. However, the authors may like to add some more discussion of the results (see Specific).

**Specific**

While I am happy with the methodology, there is (unfortunately) very little discussion of the (interesting) results. One may wonder why the authors perform (sort of) a sensitivity study with different continental set ups only to illustrate that the method is able to find equilibrium solutions. For this, one example would have been sufficient. In my view, a more thorough discussion of the results would add significant to the value of this paper.

**Answer:** Although the main focus of this manuscript was to present the method, we agree with the Reviewer that we can add further comments on the results of the different continental set ups. We have elaborated on the results and added a Discussion section in the revised manuscript. In particular, we have discussed how the parameter $\varepsilon$ effectively determines the symmetry of the system and how this affects the system dynamics in terms of stability and number of equilibria.

**Minor**

1. I may have overlooked it, but the authors may state which root searching algorithm they have used.
   **Answer:** We used a Broyden's method for finding the roots $\theta_{c_i}$ as computing the Jacobian is expensive for solutions with multiple ice lines. This is indeed not stated in the manuscript and is mentioned in the revision. We thank the Reviewer for pointing this out.

2. It seems that starting from section 2 the variable T is non-dimensional (T/Ts)) which may be confusing given the dimensional T in Eq. (1). The authors may use different symbols for dimensional and non-dimensional variables, respectively.
   **Answer:** We agree with the Reviewer that additional emphasis should be placed on the distinction between dimensional and non-dimensional temperatures. In the revised manuscript, we initially denoted non-dimensional temperatures with a tilde notation to distinguish them from the dimensional temperature T. However, dimensional temperatures appear only in the Introduction, and we believe that retaining the tilde notation throughout would reduce the readability of the manuscript. We have therefore omitted the tilde notation from Eq. (4) onward and added a statement to clarify this. We hope this clarification will prevent any confusion.

3. All equations should be numbered.
   **Answer:** Done as suggested.

**Reply to Reviewer 3**

**Answer:** We are grateful to the Reviewer for their insightful and positive feedback on our manuscript. We have thoroughly reviewed each comment and made the necessary revisions, which are outlined in detail in the responses that follow.

This interesting paper investigates the steady states of a one-dimensional energy balance model with meridional heat diffusion using a so-called boundary integral method (BIM). The novelty lies in the application of the method to the problem at hand, which allows the treatment of arbitrary discontinuity, and which allows the authors to discover multistability in the system as well as a variety of unstable steady states with several ice edges and a continent.

The paper is well-written and the results are presented clearly. I recommend publication after a few minor comments are addressed.

1. At the beginning of Section 2, it may be helpful for readers unfamiliar with the method to explicitly introduce what constitutes the boundary integral method. Either in general terms, or in the context of the presented equations.
   **Answer:** We think this is a good suggestion and have added some comments introducing the boundary integral method in general terms at the start of Section 2 in the revised manuscript.

2. For extra claricty, in Section 2.1, perhaps explicitly mention how the solutions in this work are constructed. I.e., first make an "ansatz" for the albedo function and then solve the previously obtained relations in the relevant latitudinal domains.
   **Answer:** We thank the Reviewer for suggesting the introduction of the term "ansatz," it is very appropriate in this context. We have included additional comments in Section 2.1 to clarify the application of the method. Specifically, we have stated that the BIM relies on positing an ansatz regarding the distribution of ice and water, followed by the subsequent partitioning of the domain and application of the integral identity.

3. In the Discussion, it could be helpful to put the problem of multistability into a broader context. It seems like BIM is a good tool to determine steady states analytically, and especially the possibility to find a variety of unstable states is worth mentioning. But I am wondering how the method fairs in the presence of a high number of co-existing states? It is not so clear whether Fig. 3 constitutes a "complete" bifurcation diagram, or just a continuation of several (out of potentially many more) steady states that have been found via a set of boundary conditions. Is it possible that the model shows a high degree of multistability, akin to a "fragmented" or "hierarchical" stability landscape, as has been found in other climate models (Bastiaansen et al 2022, Lohmann et al 2024)? In this case, can the BIM with its reliance on prescribing an "ansatz" for the albedo function help, or does one need to resort to numerical simulation in order to find the "global" stability landscape?
   **Answer:** The reviewer raises an important point regarding the performance of the method in the presence of a high number of co-existing states, and we have added comments on this in the "Discussion" section of the revised manuscript. We believe that the method fares very well in the presence of a high number of co-existing states, this will be further discussed below. We are confident that Fig. 3 constitutes a "complete" bifurcation diagram, particularly in terms of steady states, as this has been verified through multiple numerical simulations using a finite difference code solving the time-dependent equation. While the model does not exhibit a high degree of multistability for the configurations analyzed in this manuscript, our research (unpublished results) indicates that a large number of co-existing (unstable) states can occur in certain configurations of the North-type EBM. The BIM is capable of identifying all co-existing states in "fragmented" bifurcation structures, like e.g. in Bastiaansen, Dijkstra, and Heydt 2022, Fig. 3, for "connected" branches. We note that the BIM identifies equilibrium solutions but does not differentiate between stable and unstable states.
   Although the reviewer is correct that the method relies on prescribing an ansatz, this is generally not an issue for connected branches, i.e., branches connected to the ice-free branch (upper branch in Fig. 2) and the Snowball Earth branch (lower branch in Fig. 2) in the bifurcation diagram. Since we know that these solutions always exist for some parameter regimes, we identify the bifurcation points of these branches and then we and update the ansatz accordingly, repeating this process until the branches connect. Using this approach, all co-existing states in connected branches can be found.
   Another approach is to seek a graphical solution to the system of boundary integral equations. Posing an ansatz gives rise to distinct systems of boundary integral equations (see, e.g. Eq. (19)-(24)). These systems of equations can have multiple solutions, and a numerical method is needed to find the critical latitudes. For solutions with 1, 2, or 3 critical latitudes, the system of boundary integral equations can be solved graphically, providing a visual representation of the number of co-existing states and the location of the critical latitudes. Using this approach, one can find all co-existing states for a given ansatz with 1, 2, and 3 critical latitudes (and up to 6 if one assumes symmetry), including in isolated branches. While

prescribing all possible ansatzes to search for isolated branches can be tedious, particularly for a more complex geography, this problem is primarily one of combinatorics.

However, the graphical approach becomes impossible for more than three critical latitudes. Detecting isolated branches with more than three critical latitudes would require a different numerical approach and would generally be more challenging and expensive. We note that our work indicates that isolated branches are not common in the model. Furthermore, for the model configurations studied here, no solution with multiple critical latitudes was found to be stable. Therefore, any isolated branches with multiple critical latitudes would most likely be unstable and not detectable by numerical simulation, making the BIM the better option.

To summarize, we believe that the BIM serves as an effective framework for identifying the "global" stability landscape. Although we consider it to be somewhat hypothetical, it is worth mentioning the potential challenges associated with multiple-critical-latitude solutions in isolated branches; this is now explicitly stated in the revised manuscript. We thank the Reviewer for this insightful observation.

**References:**

- Bastiaansen et al., "Fragmented tipping in a spatially heterogeneous world", Environ. Res. Lett. 17 045006 (2022)

- Lohmann et al., "Multistability and intermediate tipping of the Atlantic Ocean circulation", Sci. Adv. 10, eadi4253 (2024)

**Reply to Reviewer 4**

**Answer:** We would like to sincerely thank the Reviewer for their thorough reading of the manuscript and for providing valuable feedback. We have carefully considered their suggestions and incorporated some of them where appropriate. However, we respectfully disagree with certain points raised, as outlined in detail below. Additionally, we find that the major revisions suggested by this Reviewer diverge significantly from those of the other Reviewers, making it challenging to incorporate all the recommendations in their entirety.

This article is concerned with zonal energy balance models (EBMs) for Earth's climate system. EBMs were first considered by Budyko and Sellers in the 1960s and studied extensively by North and coworkers in the 1970s. Needless to say, much analytical and computational work has been done since then to extend the original results to more complicated configurations. The purported novelty of the article is the application of boundary integral methods.

The configuration is kept relatively simple. The Earth is modeled as a solid sphere, symmetric with respect to the equatorial plane, with one or more zonal regions and latitudinal energy transfer (diffusion). The authors focus on steady state solutions of Eq. (1). The nondimensional form is given in Eq. (3). The equations are linear, but they allow for ice-albedo feedback.

The authors construct a Green's function for the perturbed diffusion operator in terms of Legendre polynomials. The Green's function is then used to produce solutions on the relevant subintervals (polar ice caps, land mass, and equatorial regions) and patched together to find a global representation. The main results are given in Figure 2 (bifurcation diagrams) for three slightly different configurations (distinguished by the value of the parameter $\varepsilon$). It shows that the solution of an EBM can be highly sensitive to the parameter values. Also, the bifurcation diagrams show some unexpected unstable branches.

The article is self-contained and would be a nice contribution to the literature on EBMs. The problem is that EBMs have been around for so long that a full-blown introduction with a slew of references seems like overkill. (I missed references to some articles by Ester Widiashi et al., who focused on the ice lines.) So, I recommend that the authors cut most of the introduction and give us their essential findings, leaving the technical details to the reader. After all, the problem is linear, and most readers are (or ought to be) familiar with integration by parts and linear stability analysis.

In summary: I am somewhat hesitant to recommend publication in its present form. A short summary of the new elements and findings would be acceptable.

**Answer:** Although we agree with the Reviewer that EBMs are well-known and do not require an extensive introduction, we believe that the introduction of EBMs in the manuscript is appropriately concise. It provides a brief review of previous analytical work with EBMs to highlight how the BIM fits with other analytical methods from the literature, e.g. spectral methods, for North-type models. We agree that a reference to the work of Ester Widiashi et al. is appropriate here, and this will be included in the revision. We thank the Reviewer for pointing this out.

As the manuscript introduces the BIM for solving North-type EBMs, we believe that some technical details are relevant to the

readers. We would like to clarify that the problem is not linear but piecewise linear. The nonlinearity emerges from the ice-albedo feedback, as the extent of the linear subdomains is not known and must be determined. A key novelty of the method lies in the application of boundary integral equations to address this challenge, and we believe presenting some technical details adds value to the manuscript.

**Reply to Reviewer 5**

**Answer:** We would like to sincerely thank the Reviewer for their insightful comments and positive feedback on our manuscript. We have carefully considered their suggestions and made the necessary revisions, which are detailed in the following responses.

This paper proposes the analytical method, the boundary integral method (BIM), to solve a type of simplified climate model known as a North-type energy balance model (EBM). The authors demonstrate the effectiveness of the BIM by solving the EBM with various configurations, including a zonally symmetric continent with an altered ice-albedo feedback mechanism. They show that the BIM can handle complex scenarios with multiple ice edges and ice belts, making it a powerful tool for understanding the dynamics of these simplified climate models.

    The argument is surely relevant and the methodology is well-explained, so that I recommend the publication after minor corrections. In the following, I provide some suggestions for further improving the manuscript:

1. It would be useful to provide some examples of the application and efficacy of such methodology in different research fields (e.g. how it works in the cited paleoclimatology field).
   **Answer:** We agree with the Reviewer that highlighting areas of research where the BIM can be applied, without going into too much detail, adds value to the manuscript. We address this in the latter paragraph of the "Discussion" in the revised manuscript. Potential applications in the field of paleoclimatology are discussed on lines 184 and 227.

2. Readers would appreciate knowing all the limitations of the North-type EBMs as well as the advantages.
   **Answer:** We believe that most readers are already familiar with the role of EBMs in the climate model hierarchy, as well as their limitations compared to more realistic models. Therefore, we feel that more discussion on this topic is unnecessary and respectfully decline this suggestion.

3. In my opinion, the authors should broaden the conclusions by summarizing the key points and further highlighting the importance of North-type EBMs.
   **Answer:** We agree with the Reviewer that the key points of the manuscript can be highlighted in the "Conclusions". We have therefore added a "Discussion" section to the revised manuscript and moved parts of the "Conclusions" here. In the revised "Conclusions", the key points of the method are summarized and some comments of the role of EBMs in the climate model hierarchy are also included here.

**References**

Adams, B et al. (2003). "One-dimensional daisyworld: spatial interactions and pattern formation". In: *Journal of theoretical biology* 223.4, pp. 505–513.

Alberti, T et al. (2015). "Spatial interactions in a modified Daisyworld model: Heat diffusivity and greenhouse effects". In: *Physical Review E* 92.5, p. 052717.

Alberti, Tommaso et al. (2018). "On the stability of a climate model for an Earth-like planet with land-ocean coverage". In: *Journal of Physics Communications* 2.6, p. 065018.

Bastiaansen, Robbin, Henk A Dijkstra, and Anna S von der Heydt (2022). "Fragmented tipping in a spatially heterogeneous world". In: *Environmental Research Letters* 17.4, p. 045006.

Fluteau, Frederic (2003). "Earth dynamics and climate changes". In: *Comptes Rendus Geoscience* 335.1, pp. 157–174.

Hsiao, George C and Wolfgang L Wendland (2008). *Boundary integral equations*. Springer.

Morino, Luigi and Renzo Piva (2012). *Boundary Integral Methods: Theory and Applications*. Springer Science & Business Media.

Rombouts, Jan and M Ghil (2015). "Oscillations in a simple climate–vegetation model". In: *Nonlinear Processes in Geophysics* 22.3, pp. 275–288.

Walsh, James and Esther Widiasih (2014). "A DYNAMICS APPROACH TO A LOW-ORDER CLIMATE MODEL." In: *Discrete & Continuous Dynamical Systems-Series B* 19.1.

Widiasih, Esther R (2013). "Dynamics of the Budyko energy balance model". In: *SIAM Journal on Applied Dynamical Systems* 12.4, pp. 2068–2092.

Widiasih, Esther R, Andrew Keane, and Malte F Stuecker (2024). "The Mid-Pleistocene Transition from Budyko's Energy Balance Model". In: *Physica D: Nonlinear Phenomena* 458, p. 133991.